# `ST-MAML` : A Stochastic-Task based Method for Task-Heterogeneous Meta-Learning

**Zhe Wang**[1]     **Jake Grigsby**[1]     **Arshdeep Sekhon**[1]     **Yanjun Qi**[1]

[1]Computer Science Dept., University of Virginia, Charlottesville, Virginia, USA

## Abstract

Optimization-based meta-learning typically assumes tasks are sampled from a single distribution – an assumption that oversimplifies and limits the diversity of tasks that meta-learning can model. Handling tasks from multiple distributions is challenging for meta-learning because it adds ambiguity to task identities. This paper proposes a novel method, `ST-MAML`, that empowers model-agnostic meta-learning (`MAML`) to learn from multiple task distributions. `ST-MAML` encodes tasks using a stochastic neural network module, that summarizes every task with a stochastic representation. The proposed Stochastic Task (`ST`) strategy learns a distribution of solutions for an ambiguous task and allows a meta-model to self-adapt to the current task. `ST-MAML` also propagates the task representation to enhance input variable encodings. Empirically, we demonstrate that `ST-MAML` outperforms the state-of-the-art on two few-shot image classification tasks, one curve regression benchmark, one image completion problem, and a real-world temperature prediction application.

## 1 INTRODUCTION

Meta-learning aims to train a model on multiple machine learning tasks to adapt to a new task with only a few training samples. Optimization-based meta-learning like model-agnostic meta-learning (MAML) facilitate such a goal by involving the optimization process. For example, MAML trains a global initialization of model parameters that are close to the optimal parameter values of every task (Finn et al., 2017). Recent methods expand MAML's "global initialization" to a notion of "globally shared knowledge", including not only initialization (Finn et al., 2017; Li et al., 2017; Rajeswaran et al., 2019) but also update rules (Andrychowicz et al., 2016; Ravi and Larochelle,

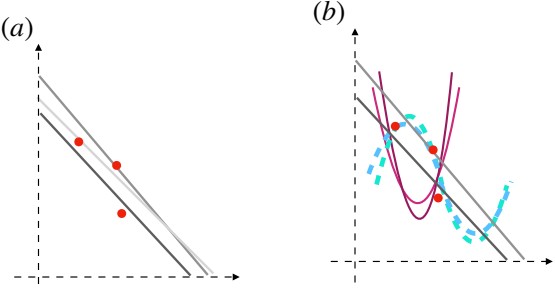

Figure 1: We are given three red dots representing the training data for a meta-test task. The dashed and solid curves are potential explanations of the data (better read in color). (a) Homogeneous setup. All meta-training tasks are sampled from linear regression family. (b) Heterogeneous setup. The meta-training tasks are sampled from three possible function families including *sinusoids, straight line, and quadratic*. It is difficult to figure out what family this meta-test task is sampled from, due to limited annotated data and three possible meta distributions.

2017). Globally shared knowledge allows these methods to produce good generalization performance on new tasks with a small number of training samples.

Most optimization-based meta-learning algorithms assume all tasks $\mathcal{T}$ are identically and independently sampled from a single distribution (Andrychowicz et al., 2016; Finn et al., 2017; Li et al., 2017; Ravi and Larochelle, 2017; Rusu et al., 2018). We refer to meta-learning's task distribution as the "meta-distribution". Formally, these methods assume $\mathcal{T} \sim P(\mathcal{T})$. Real-world tasks, however, may come from multiple meta-distributions, $\mathcal{T} \sim \{P_1(\mathcal{T}), P_2(\mathcal{T}), \cdots, P_k(\mathcal{T})\}$. For instance, when analyzing multiple writers' hand written digits, writers from different age group (like children versus adults) indicate different meta-distributions. This more challenging setup, we call task heterogeneity, poses technical challenges to homogenous strategies like MAML (Vuorio et al., 2019).

For task heterogeneity, a naive and widely accepted meta-learning solution first learns a globally shared initialization across all meta-distributions and then tailors model parameters to the current task (Vuorio et al., 2019; Yao et al.,

*Accepted for the 38th Conference on Uncertainty in Artificial Intelligence* (UAI 2022).

2020, 2019; Lee and Choi, 2018; Oreshkin et al., 2018). The tailoring step needs to rely on the task-specific information or, ideally, the identity information of the task. It, therefore, requires the meta-learner to infer the potential identity of a new task from a limited number of annotated samples (Finn et al., 2018). This requirement raises severe uncertainty issues – a challenge known as "task ambiguity". Figure 1 provides a concrete example of the task ambiguity that arises from limited annotated data and unknown meta distribution when facing task heterogeneity. Surprisingly, recent optimization-based meta-learning literature pays little attention to the task ambiguity challenge (Vuorio et al., 2019; Yao et al., 2020, 2019; Lee and Choi, 2018). Besides, the task heterogeneity amplifies the "distribution shift" issue (Zhang et al., 2021; Dubey et al., 2021). The difference between two tasks can significantly increase in the heterogeneous setup since tasks are from various meta-distributions. This paper proposes a novel meta-learning method `ST-MAML` for the task heterogeneity challenge. Our approach extends MAML by modeling tasks as a stochastic variable that we call the "stochastic task". Stochastic tasks (STs) let us learn a distribution of solutions to capture the uncertainty of an ambiguous new task. At the same time, STs enable self-adaptive model initialization based on the current task. We use variational inference as a solver and the whole learning process is meta-distribution agnostic. We apply `ST-MAML` to a wide range of common meta-learning benchmarks including synthetic regression, image completion, and few-shot image classification, where `ST-MAML` exceeds the performance of existing work. We also build a large temperature prediction dataset that highlights the challenges of real-world meta-distributions. Our empirical results demonstrate that `ST-MAML` outperforms the MAML baselines by 40% on this new task.

## 2 METHODS

### 2.1 PRELIMINARIES ON META LEARNING

We describe a supervised learning task in meta-learning as

$$\mathcal{T} = \{\mathcal{L}oss(), \boldsymbol{f}_{\boldsymbol{\theta}_{\mathcal{T}}}, \boldsymbol{D}_{\mathcal{T}}^{tr}, \boldsymbol{D}_{\mathcal{T}}^{te}\}$$
$$= \{\mathcal{L}oss(), \boldsymbol{f}_{\boldsymbol{\theta}_{\mathcal{T}}}, [\boldsymbol{X}_{\mathcal{T}}^{tr}, \boldsymbol{Y}_{\mathcal{T}}^{tr}], [\boldsymbol{X}_{\mathcal{T}}^{te}, \boldsymbol{Y}_{\mathcal{T}}^{te}]\}, \quad (1)$$

Here $\mathcal{L}oss()$, which takes as input model $\boldsymbol{f}_{\boldsymbol{\theta}_{\mathcal{T}}}$ and dataset, describes the loss function that measures the quality of learner $\boldsymbol{f}_{\boldsymbol{\theta}_{\mathcal{T}}}$, whose parameter weight is $\boldsymbol{\theta}_{\mathcal{T}}$. Every task includes an annotated training set $\boldsymbol{D}_{\mathcal{T}}^{tr} = [\boldsymbol{X}_{\mathcal{T}}^{tr}, \boldsymbol{Y}_{\mathcal{T}}^{tr}]$ and a test set $\boldsymbol{D}_{\mathcal{T}}^{te} = [\boldsymbol{X}_{\mathcal{T}}^{te}, \boldsymbol{Y}_{\mathcal{T}}^{te}]$. During meta-training, the test set $\boldsymbol{D}_{\mathcal{T}}^{te}$ is fully observed, but during meta-testing only its input $\boldsymbol{X}_{\mathcal{T}}^{te}$ is available. $\boldsymbol{D}_{\mathcal{T}}^{tr}$ and $\boldsymbol{D}_{\mathcal{T}}^{te}$ are sampled from $\mathcal{X} \times \mathcal{Y}$, $\mathcal{X}$ describes the input space and $\mathcal{Y}$ is the output space.

The goal of meta learning is to train a learning machine which can perform well on $\boldsymbol{D}_{\mathcal{T}}^{te}$ after fine-tuning on this task's training set $\boldsymbol{D}_{\mathcal{T}}^{tr}$. The difficulty lies at finding a balance between underfitting to all tasks and overfitting to any particular task. MAML (Finn et al., 2017) achieves such

a goal by learning a globally shared weight initialization $\boldsymbol{\theta}^*$ that is close to the optimal weight parameter of every task. We can write its training objective for getting the best initialization $\boldsymbol{\theta}^*$ as:

$$\min_{\boldsymbol{\theta}} \underset{\mathcal{T} \sim P(\mathcal{T})}{\mathbf{E}} [\mathcal{L}oss(\boldsymbol{f}_{\boldsymbol{\theta}_{\mathcal{T}}^1}, \boldsymbol{D}_{\mathcal{T}}^{te})],$$
$$\text{where} \quad \boldsymbol{\theta}_{\mathcal{T}}^1 = \boldsymbol{\theta}_{\mathcal{T}}^0 - \alpha \nabla_{\boldsymbol{\theta}} [\mathcal{L}oss(f_{\boldsymbol{\theta}_{\mathcal{T}}^0}, D_{\mathcal{T}}^{tr})],$$
$$\text{and} \quad \boldsymbol{\theta}_{\mathcal{T}}^0 = \boldsymbol{\theta}. \quad (2)$$

MAML samples a set of tasks $\{\mathcal{T}\}$ from the meta distribution $P(\mathcal{T})$ and initialize each task's weight $\boldsymbol{\theta}_{\mathcal{T}}^0$ from the global knowledge $\boldsymbol{\theta}$ (to be learnt): i.e., setting $\boldsymbol{\theta}_{\mathcal{T}}^0 = \boldsymbol{\theta}$. On each task, the learner performs gradient descent on its training set $\boldsymbol{D}_{\mathcal{T}}^{tr}$ to reach task-specific fine-tuned parameters $\boldsymbol{\theta}_{\mathcal{T}}^1$. The test set $\boldsymbol{D}_{\mathcal{T}}^{te}$ of task $\mathcal{T}$ is used for evaluating parameter $\boldsymbol{\theta}_{\mathcal{T}}^1$, and the evaluation will be used as the objective to optimize for learning the best global knowledge $\boldsymbol{\theta}$.

In probabilistic language, the above objective (in Eq. (2)) can be equivalently framed as maximizing the likelihood:

$$\max_{\boldsymbol{\theta}} \prod_{\mathcal{T} \sim P(\mathcal{T})} [\mathcal{L}(\mathcal{T})] = \prod_{\mathcal{T} \sim P(\mathcal{T})} p(\boldsymbol{Y}_{\mathcal{T}}^{te} | \boldsymbol{X}_{\mathcal{T}}^{te}, \boldsymbol{D}_{\mathcal{T}}^{tr}, \boldsymbol{\theta}) \quad (3)$$

$$= \prod_{\mathcal{T} \sim P(\mathcal{T})} \sum_{\boldsymbol{\theta}_{\mathcal{T}}^1} p(\boldsymbol{Y}_{\mathcal{T}}^{te} | \boldsymbol{X}_{\mathcal{T}}^{te}, \boldsymbol{\theta}_{\mathcal{T}}^1) p(\boldsymbol{\theta}_{\mathcal{T}}^1 | \boldsymbol{D}_{\mathcal{T}}^{tr}, \boldsymbol{\theta}), \quad (4)$$

where $p(\boldsymbol{\theta}_{\mathcal{T}}^1 | \boldsymbol{D}_{\mathcal{T}}^{tr}, \boldsymbol{\theta})$ is a Dirac distribution derived by minimizing the negative log-likelihood(NLL) on $\boldsymbol{D}_{\mathcal{T}}^{tr}$ with gradient descent.

### 2.2 PREVIOUS HETEROGENEOUS META LEARNING

Task-homogeneous meta-learning assumes that there exists one meta-distribution $P(\mathcal{T})$ and all tasks are identically and independently (i.i.d.) sampled from $P(\mathcal{T})$. Differently, in a task-heterogeneous setup, there exist multiple meta-distributions $\mathcal{T} \sim \{P_1(\mathcal{T}), P_2(\mathcal{T}), \cdots, P_k(\mathcal{T})\}$.

We can naively use MAML and assign all tasks with the same global initialization (though they come from different distributions). Figure 1(a, b) show that the "task ambiguity" issue is more critical in task-heterogeneous setup and will hinder the generalization from MAML initialization since multiple very different task distributions exist.

A handful of previous works learn a customized initialization that was tailored from global initialization, in order to tackle the task heterogeneity challenge. MMAML (Vuorio et al., 2019) learns a deterministic task embedding with an RNN module. HSML (Yao et al., 2019) manually designs a task clustering algorithm to assign tasks to different clusters, then customizes the global initialization to each cluster. ARML (Yao et al., 2020) models global knowledge and task-specific knowledge as graphs; the interaction between tasks is modeled by message passing.

However, none of the recent works consider the task ambiguity issue when solving task-heterogeneous domains. Most frameworks are still based on the assumption that only one

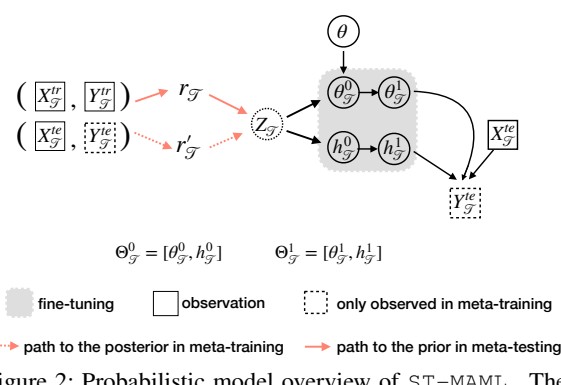

$$\Theta_{\mathcal{T}}^0 = [\theta_{\mathcal{T}}^0, h_{\mathcal{T}}^0] \qquad \Theta_{\mathcal{T}}^1 = [\theta_{\mathcal{T}}^1, h_{\mathcal{T}}^1]$$

▨ fine-tuning    ☐ observation    ⬚ only observed in meta-training

···▷ path to the posterior in meta-training    ⟶ path to the prior in meta-testing

Figure 2: Probabilistic model overview of ST-MAML . The stochastic variable $Z_{\mathcal{T}}$ conditioned on task information $(X_{\mathcal{T}}, Y_{\mathcal{T}})$ is used for model's self-adaptation and input variable's re-encoding.

---

**Algorithm 1** ST-MAML META-TRAINING PROCEDURE.

1: **Input:** Meta-distribution set $\{P_1(\mathcal{T}), \cdots, P_k(\mathcal{T})\}$, Hyper-parameters $\gamma_1$ and $\gamma_2$.
2: Randomly initialize model parameter $\boldsymbol{\theta}$, stochastic task module parameters $\boldsymbol{\phi}$, tailoring module parameters $\boldsymbol{w}$, input encoding parameters $\boldsymbol{\beta}$.
3: **while** not DONE **do**
4:   Sample batches of $m$ tasks $\{\mathcal{T}\}$ from meta-distribution set.
5:   **for** every task $\mathcal{T}$ **do**
6:     Infer the posterior distribution of stochastic task variable $q(\boldsymbol{Z}_{\mathcal{T}}|\mathcal{T})$ and sample $\boldsymbol{z}_{\mathcal{T}} \sim q(\boldsymbol{Z}_{\mathcal{T}}|\mathcal{T})$. [eq.(8) and eq.(10)]
7:     Tailor $\boldsymbol{\theta}$ with sample $\boldsymbol{z}_{\mathcal{T}}$ to get task-specific initialization $\boldsymbol{\theta}_{\mathcal{T}}^0$. [eq.(12)]
8:     Revise the encoding of input variable by augmenting the raw input. [eq.(13)]
9:     Evaluate the inner loss $\mathcal{L}_{in}(\mathcal{T})$ on training set $\boldsymbol{D}_{\mathcal{T}}^{tr}$. [eq.(17)]
10:    Compute adapted parameter and augmented feature with gradient descent [eq.(18)]:
       $\boldsymbol{\theta}_{\mathcal{T}}^1 = \boldsymbol{\theta}_{\mathcal{T}}^0 - \gamma_1 \nabla_{\boldsymbol{\theta}_{\mathcal{T}}^0} \mathcal{L}_{in}(\mathcal{T}), \mathbf{h}_{\mathcal{T}}^1 = \mathbf{h}_{\mathcal{T}}^0 - \gamma_1 \nabla_{\mathbf{h}_{\mathcal{T}}^0} \mathcal{L}_{in}(\mathcal{T}).$
11:   **end for**
12:   Update $\boldsymbol{\theta}, \boldsymbol{\phi}, \boldsymbol{w}, \boldsymbol{\beta}$ with $\gamma_2 \frac{1}{m} \nabla_{[\boldsymbol{\theta},\boldsymbol{\phi},\boldsymbol{w},\boldsymbol{\beta}]} \sum_{\mathcal{T}} \mathcal{L}_{ELBO}(\mathcal{T})$. [eq.(16)]
13: **end while**

---

distribution exists to explain a task's observed training set (e.g., a new task should be assigned to only one cluster in HSML). However, the source of a task can be highly uncertain based on limited annotated data. Figure 1(b) shows that there can be multiple explanations of an observed dataset in the task-heterogeneous setup and we should not expect to obtain a unique solution.

### 2.3 STOCHASTIC $Z_{\mathcal{T}}$ TO ENCODE TASK

When facing the task-heterogeneous setup, we hypothesize that a meta-learner that can encode potential tasks' patterns will help alleviate the task ambiguity issue. These patterns could describe valuable information about tasks like the more possible shapes of curves for a regression meta-application. Moreover, we propose to enable task encoding with uncertainty estimates. This is because learning a task representation from its limited annotated data is challenging and such uncertainty measures can help inform the downstream meta-adaptation to new tasks (see Figure 1(b)). This hypothesis motivates us to describe a task $\mathcal{T}$ with a stochastic variable $Z_{\mathcal{T}}$ and model its distribution to condition on observations. With this additional latent variable, we can rewrite the per task likelihood $\mathcal{L}(\mathcal{T})$ in Eq. (3) as:

$$\mathcal{L}(\mathcal{T}) = \sum_{Z_{\mathcal{T}}} p(Y_{\mathcal{T}}^{te}|X_{\mathcal{T}}^{te}, \boldsymbol{D}_{\mathcal{T}}^{tr}, \boldsymbol{Z}_{\mathcal{T}}, \boldsymbol{\theta}) p(\boldsymbol{Z}_{\mathcal{T}}|\boldsymbol{D}_{\mathcal{T}}^{tr}). \quad (5)$$

We assume in the second term from above, $Z_{\mathcal{T}}$ only conditions on $\boldsymbol{D}_{\mathcal{T}}^{tr}$. Figure 2 shows our design.

In Section (2.5), we show that the likelihood is intractable as defined above, and choose to maximize its evidence lower bound (a.k.a ELBO) instead. Optimizing this variational objective requires the prior $p(Z_{\mathcal{T}}|\boldsymbol{D}_{\mathcal{T}}^{tr})$ and the posterior $q(Z_{\mathcal{T}}|\mathcal{T})$. We model the prior $p(Z_{\mathcal{T}}|\boldsymbol{D}_{\mathcal{T}}^{tr})$ as a Gaussian distribution, whose mean and variance are outputs from a multi-layer perceptron (MLP) module with input vector $r_{\mathcal{T}}$:

$$p(\boldsymbol{Z}_{\mathcal{T}}|\boldsymbol{D}_{\mathcal{T}}^{tr}) = \mathcal{N}(\boldsymbol{\mu}(r_{\mathcal{T}}), \boldsymbol{\sigma}(r_{\mathcal{T}})). \quad (6)$$

Here vector $r_{\mathcal{T}}$ is a vector summarizing the encoding of a task $\mathcal{T}$. We propose a neural network module to learn $r_{\mathcal{T}}$ from the sample observations $\boldsymbol{D}_{\mathcal{T}}^{tr}$. The training observations of task $\mathcal{T}$ consist of unordered annotated data pairs $[(\mathbf{x}_{\mathcal{T}}^{tr}, \mathbf{y}_{\mathcal{T}}^{tr})]$. Permutation invariance is a desirable property for functions acting on sets. Zaheer et al. (2017) showed any function acting on sets $S$ is permutation invariant if and only if it can be decomposed as $\rho(\sum_{\mathbf{s} \in S} \phi(\mathbf{s}))$ for suitable choice of transformations $\rho, \phi$. We follow such a design, and encode a task by encoding every pair of its observation in $\boldsymbol{D}_{\mathcal{T}}^{tr}$ through a neural network layer:

$$r_{\mathcal{T},j} = \boldsymbol{g}_{\boldsymbol{\phi}}^{Enc}(\mathbf{x}_{\mathcal{T},j}^{tr}, \mathbf{y}_{\mathcal{T},j}^{tr}), \quad j = 1, \cdots, |\boldsymbol{D}_{\mathcal{T}}^{tr}|, \quad (7)$$

$$r_{\mathcal{T}} = \frac{1}{|\boldsymbol{D}_{\mathcal{T}}^{tr}|} \sum_{j=1}^{|\boldsymbol{D}_{\mathcal{T}}^{tr}|} r_{\mathcal{T},j}. \quad (8)$$

Eq. (8) uses average function as aggregation operator to obtain the task embedding because it is able to remove the inductive bias due to different sizes of training set from $r_{\mathcal{T}}$. In Eq. (7), $\boldsymbol{g}_{\boldsymbol{\phi}}^{Enc}()$ is implemented as a MLP module with learnable parameter $\phi$.

We then approximate the intractable posterior distribution $q(Z_{\mathcal{T}}|\mathcal{T})$ of $Z_{\mathcal{T}}$ as conditioned on the whole $\{\boldsymbol{D}_{\mathcal{T}}^{tr}, \boldsymbol{D}_{\mathcal{T}}^{te}\}$ (see Appendix S2):

$$q(\boldsymbol{Z}_{\mathcal{T}}|\mathcal{T}) = q(\boldsymbol{Z}_{\mathcal{T}}|\boldsymbol{D}_{\mathcal{T}}^{tr}, \boldsymbol{D}_{\mathcal{T}}^{te}) = \mathcal{N}(\boldsymbol{\mu}(r_{\mathcal{T}}'), \boldsymbol{\sigma}(r_{\mathcal{T}}')), \quad (9)$$

$$r_{\mathcal{T}}' = \frac{1}{|\mathcal{T}|} \sum_{j=1}^{|\mathcal{T}|} r_{\mathcal{T},j}, \quad j = 1, \cdots, (|\boldsymbol{D}_{\mathcal{T}}^{tr}| + |\boldsymbol{D}_{\mathcal{T}}^{te}|), \quad (10)$$

where $|\mathcal{T}| = |\boldsymbol{D}_{\mathcal{T}}^{tr}| + |\boldsymbol{D}_{\mathcal{T}}^{te}|$ , $\boldsymbol{\mu}(\cdot)$ and $\boldsymbol{\sigma}(\cdot)$ are the same MLP modules we have in Eq. (6).

### 2.4 ST-MAML : SELF ADAPTATION WITH $Z_{\mathcal{T}}$

We propose to revise MAML for the heterogeneous meta-learning setup using the summary task representation $Z_{\mathcal{T}}$, creating ST-MAML . $Z_{\mathcal{T}}$ helps tailor the global initialization

$\boldsymbol{\theta}$ to task-specific initialization $\boldsymbol{\theta}_{\mathcal{T}}^0$ for a task $\mathcal{T}$. Its basic motivation is to improves flexibility by incorporating task information into the model. This self adaption design is motivated by the recent ideas that design self-adaptation conditioned on global knowledge to conquer distribution shift issue in domain generalization/adaptation (Zhang et al., 2021; Dubey et al., 2021; Xiao et al., 2021; Vuorio et al., 2019).

There exist many potential ways to use $\boldsymbol{Z}_{\mathcal{T}}$ to tailor the global initialization $\boldsymbol{\theta}$ to task-specific initialization $\boldsymbol{\theta}_{\mathcal{T}}^0$. We assume our target learning machine is a composition of a base learner and a task learner:

$$\boldsymbol{f}_{\boldsymbol{\theta}_{\mathcal{T}}} = \boldsymbol{f}_{\boldsymbol{\theta}_c}(\boldsymbol{f}_{\boldsymbol{\theta}_b}).$$

Here the base learner's parameters are $\boldsymbol{\theta}_b$, and the task learner's parameters are $\boldsymbol{\theta}_c$. For example, in an image classification domain the base learner would be the the the CNN backbone and the task learner would be the last linear layer. We can then rewrite $\boldsymbol{\theta} = [\boldsymbol{\theta}_b, \boldsymbol{\theta}_c]$. We propose to only customize $\boldsymbol{\theta}_c$ with $\boldsymbol{Z}_{\mathcal{T}}$:

$$\boldsymbol{\theta}_{\mathcal{T}}^0 = g_{\boldsymbol{w}}^{Gate}(\boldsymbol{\theta}, \boldsymbol{Z}_{\mathcal{T}}) = [\boldsymbol{\theta}_b, \sigma(\boldsymbol{w_1}\boldsymbol{z}_{\mathcal{T}} + \boldsymbol{w}_0) \odot \boldsymbol{\theta}_c], \quad (11)$$
$$= [\boldsymbol{\theta}_b, \sigma(\boldsymbol{w}_{gate}) \odot \boldsymbol{\theta}_c] \quad (12)$$

Here $\boldsymbol{z}_{\mathcal{T}}$ is sampled from the distribution $q(\boldsymbol{Z}_{\mathcal{T}}|\mathcal{T})$ during meta-training and from $p(\boldsymbol{Z}_{\mathcal{T}}|\boldsymbol{D}_{\mathcal{T}}^{tr})$ during meta-testing. $\sigma$ is the sigmoid function, $\odot$ represents the element-wise multiplication, $\boldsymbol{w} = [\boldsymbol{w}_1, \boldsymbol{w}_0]^T$ are learnable parameters. $\boldsymbol{w}_{gate}$, the gate vector will apply element-wise scaling to navigate global initialization $\boldsymbol{\theta}$ to task-specific initialization $\boldsymbol{\theta}_{\mathcal{T}}^0$.

Moreover, we design additional customized knowledge for task $\mathcal{T}$. The basic intuition is that the final prediction of a meta-learner depends on both model parameters and input representations. To increase the capacity of the task-specific knowledge, we propose to further propagate task representation $\boldsymbol{Z}_{\mathcal{T}}$ into encoding augmented feature representations we denote as $\mathbf{h}_{\mathcal{T}}$. We concatenate $\mathbf{h}_{\mathcal{T}}$ with a sample's input representation $\mathbf{x}_{\mathcal{T}}$, and feed the combined vector $\hat{\mathbf{x}}_{\mathcal{T}}$ to our learning machine as its new input.

$$\mathbf{h}_{\mathcal{T}}^0 = g_{\boldsymbol{\beta}}^{In}(\boldsymbol{Z}_{\mathcal{T}}) = \boldsymbol{\beta}_1 \boldsymbol{z}_{\mathcal{T}} + \boldsymbol{\beta}_0, \quad \hat{\mathbf{x}}_{\mathcal{T}} = [\mathbf{x}_{\mathcal{T}}, \mathbf{h}_{\mathcal{T}}^0]. \quad (13)$$

Same as Eq. (12), $\boldsymbol{z}_{\mathcal{T}}$ is sampled from its distribution, $\boldsymbol{\beta} = [\boldsymbol{\beta}_1, \boldsymbol{\beta}_0]$ are learnable parameters.

Now when facing a new task $\mathcal{T}$, a meta-model will first generate the task-specific knowledge that includes both augmented feature $\mathbf{h}_{\mathcal{T}}$ and task-specific parameter $\boldsymbol{\theta}_{\mathcal{T}}$. We denote the combined knowledge set for task $\mathcal{T}$ as:

$$\boldsymbol{\Theta}_{\mathcal{T}} = [\boldsymbol{\theta}_{\mathcal{T}}, \mathbf{h}_{\mathcal{T}}]. \quad (14)$$

This is the meta-knowledge we need to learn in ST−MAML . We note its initial values as $\boldsymbol{\Theta}_{\mathcal{T}}^0 = [\boldsymbol{\theta}_{\mathcal{T}}^0, \mathbf{h}_{\mathcal{T}}^0]$ and fine-tuned values as $\boldsymbol{\Theta}_{\mathcal{T}}^1 = [\boldsymbol{\theta}_{\mathcal{T}}^1, \mathbf{h}_{\mathcal{T}}^1]$.

Aiming to learn the meta knowledge defined in Eq. (14), we

can rewrite our objective (task likelihood) in Eq. (5):

$$\mathcal{L}(\mathcal{T}) = \sum_{\boldsymbol{\Theta}_{\mathcal{T}}^0, \boldsymbol{\Theta}_{\mathcal{T}}^1, \boldsymbol{z}_{\mathcal{T}}} p(\boldsymbol{Y}_{\mathcal{T}}^{te}|\boldsymbol{X}_{\mathcal{T}}^{te}, \boldsymbol{\Theta}_{\mathcal{T}}^1) p(\boldsymbol{\Theta}_{\mathcal{T}}^1|\boldsymbol{\Theta}_{\mathcal{T}}^0, \boldsymbol{D}_{\mathcal{T}}^{tr})$$
$$p(\boldsymbol{\theta}_{\mathcal{T}}^0|\boldsymbol{\theta}, \boldsymbol{Z}_{\mathcal{T}}) p(\mathbf{h}_{\mathcal{T}}^0|\boldsymbol{Z}_{\mathcal{T}}) p(\boldsymbol{Z}_{\mathcal{T}}|\boldsymbol{D}_{\mathcal{T}}^{tr}). \quad (15)$$

This follows the Bayesian graph provided in Figure 2.

**Design Choices:** There exist many other possible probabilistic designs besides Figure 2. For instance, we can model every variable in the figure as a stochastic distribution and build a complicated hybrid framework. However, this will lead to excessive stochasticity and increase the potential of underfitting in a limited data situation. Instead, similar to $p(\boldsymbol{\Theta}_{\mathcal{T}}^1|\boldsymbol{\Theta}_{\mathcal{T}}^0, \boldsymbol{D}_{\mathcal{T}}^{tr})$, we choose to model both $p(\mathbf{h}_{\mathcal{T}}^0|\boldsymbol{Z}_{\mathcal{T}})$ and $p(\boldsymbol{\theta}_{\mathcal{T}}^0|\boldsymbol{\theta}, \boldsymbol{Z}_{\mathcal{T}})$ as deterministic (see Eq. (12) and Eq. (13)) that allow us to employ an amortized variational inference technique Ravi and Beatson (2019).

Our design is different from recent probabilistic extensions of MAML Finn et al. (2018); Yoon et al. (2018). They conduct inference on model parameters $\boldsymbol{\theta}_{\mathcal{T}}$ (initial value $\boldsymbol{\theta}_{\mathcal{T}}^0$ or fine-tuned value $\boldsymbol{\theta}_{\mathcal{T}}^1$). Our ST−MAML shifts the burden of variational inference to the task representation $\boldsymbol{Z}_{\mathcal{T}}$, whose dimension is of multiple orders smaller than the size of model parameters.

## 2.5  ST−MAML : UPDATE RULES

**Variational Objective:** To optimize the intractable likelihood as defined in Eq. (15), we choose to maximize its evidence lower bound (a.k.a ELBO) instead:

$$\mathcal{L}_{ELBO}(\mathcal{T}) = \mathbf{E}_{\boldsymbol{\Theta}_{\mathcal{T}}^1 \sim q(\boldsymbol{\Theta}_{\mathcal{T}}^1|\mathcal{T})} \log p(\boldsymbol{Y}_{\mathcal{T}}^{te}|\boldsymbol{X}_{\mathcal{T}}^{te}, \boldsymbol{\Theta}_{\mathcal{T}}^1)$$
$$- KL(q(\boldsymbol{Z}_{\mathcal{T}}|\mathcal{T})||p(\boldsymbol{Z}_{\mathcal{T}}|\boldsymbol{D}_{\mathcal{T}}^{tr})). \quad (16)$$

During meta-training, we sample $m$ tasks and optimize the empirical average $\frac{1}{m} \sum_{t=1}^{m} \mathcal{L}_{ELBO}(\mathcal{T}_t)$.

**Update Rules:** Much like MAML, the optimization of ST−MAML consists of two nested loops. Figure 3 shows the iterative optimization process. In the inner loop, for the $j_{th}$ training data, we concatenate $\mathbf{x}_{\mathcal{T},j}^{tr}$ with augmented feature $\mathbf{h}_{\mathcal{T}}^0$ to get augmented input vector $\hat{\boldsymbol{x}}_{\mathcal{T},j}^{tr}$. We feed $\hat{\boldsymbol{x}}_{\mathcal{T},j}^{tr}$ into the learning machine $\boldsymbol{f}$ whose parameter is $\boldsymbol{\theta}_{\mathcal{T}}^0$ to calculate the inner loss:

$$\mathcal{L}_{in}(\mathcal{T}) = \frac{1}{|\boldsymbol{D}_{\mathcal{T}}^{tr}|} \sum_{j=1}^{|\boldsymbol{D}_{\mathcal{T}}^{tr}|} \mathcal{L}(\boldsymbol{f}_{\boldsymbol{\theta}_{\mathcal{T}}^0}, [\hat{\mathbf{x}}_{\mathcal{T},j}^{tr}, \mathbf{y}_{\mathcal{T},j}^{tr}]). \quad (17)$$

The inner loss is then used for updating $\boldsymbol{\theta}_{\mathcal{T}}^0$ and $h_{\mathcal{T}}^0$:

$$\mathbf{h}_{\mathcal{T}}^1 = \mathbf{h}_{\mathcal{T}}^0 - \frac{\partial \mathcal{L}_{in}(\mathcal{T})}{\partial \mathbf{h}_{\mathcal{T}}^0}, \quad \boldsymbol{\theta}_{\mathcal{T}}^1 = \boldsymbol{\theta}_{\mathcal{T}}^0 - \frac{\partial \mathcal{L}_{in}(\mathcal{T})}{\partial \boldsymbol{\theta}_{\mathcal{T}}^0}. \quad (18)$$

Figure 3 shows we can optimize the inner loss for $K$ iterations to achieve a closer approximation for optimal values in Eq. (17). In the outer loop, we maximize the approximated ELBO $\mathcal{L}_{ELBO}$ in Eq. (16) using a batch of $m$

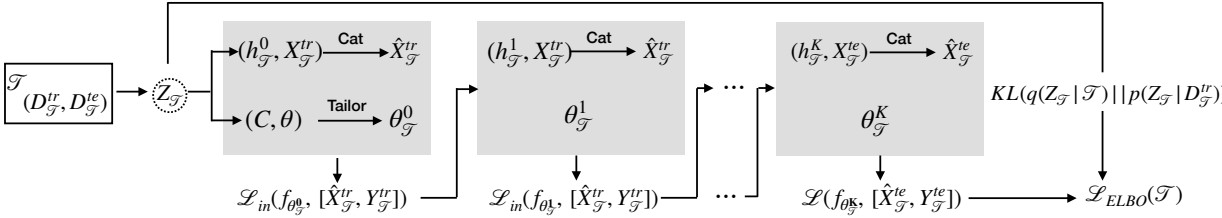

Figure 3: Iterative optimization process. In the inner loop, Starting from task-specific parameter initialization $\boldsymbol{\theta}_{\mathcal{T}}^0$ and augmented features $\mathbf{h}_{\mathcal{T}}^0$, their fine-tuned values $\boldsymbol{\theta}_{\mathcal{T}}^K$, $\mathbf{h}_{\mathcal{T}}^K$ are inferred by performing gradient descent on the training set $\boldsymbol{D}_{\mathcal{T}}^{tr}$ for $K$ iterations.

tasks. The amortized variational technique allows us to conduct the sampling from $q(\boldsymbol{\Theta}_{\mathcal{T}}^1|\mathcal{T})$ by first sampling from $q(\boldsymbol{Z}_{\mathcal{T}}|\mathcal{T})$ and then applying a deterministic transformation using Eq. (12) and Eq. (13).

**Algorithm of `ST-MAML` :** We describe the procedure of `ST-MAML` in the form of pseudo code as shown in Algorithm 1. Note, parameters of neural functions $\boldsymbol{\mu}(\cdot)$, $\boldsymbol{\sigma}(\cdot)$, $\boldsymbol{g}_{\boldsymbol{\phi}}^{Enc}()$, $\boldsymbol{g}_{\boldsymbol{w}}^{Gate}()$, and $\boldsymbol{g}_{\boldsymbol{\beta}}^{In}()$ are updated in the outer loop.

**Theoretical Analysis of `ST-MAML` :** We also provide a second interpretation of our objective from an information bottleneck perspective and prove they lead to exactly the same target. See Appendix S3 for detailed proofs.

### 2.6 CONNECTING TO RELATED WORK

Optimization-based meta-learning methods facilitate the model's adaption to new tasks through global knowledge learned by the optimization process. Meta-LSTM (Ravi and Larochelle, 2017) meta-learns the update rule with an RNN meta-learner. MAML (Finn et al., 2017) trains a global initialization close to the optimal value of every task. Leveraging diverse meta-knowledge further accelerates the learning process. In Meta-SGD (Li et al., 2017), the meta-knowledge consists of both initialization and learning rate. ALFA (Baik et al., 2020) proposes to meta-learn both initialization and hyperparameter update module. Most methods assign the same global knowledge to every task that leads to sub-optimal solutions for heterogeneous settings. Besides, they are all deterministic and can only learn one solution for a new task.

Bayesian approaches are a long-standing discipline that incorporates uncertainty in modeling. Multiple recent works extend MAML into the Bayesian framework and recast meta-learning as the probabilistic framework (Finn et al., 2018; Grant et al., 2018; Yoon et al., 2018; Ravi and Beatson, 2019; Garnelo et al., 2018b). BMAML (Yoon et al., 2018) recast MAML into probabilistic framework and provides a Bayesian explanation of MAML. PLATIPUS (Finn et al., 2018) builds upon amortized variational inference and injects Gaussian noise into the gradient during the meta-testing time to learn a distribution over model parameters. LLAMA (Grant et al., 2018) applies Laplace approximation for modeling the parameter distribution, but it requires the approximation of a high dimensional covariance ma-

trix. These methods view model parameters (i.e. network weights and bias) as random variables and perform inference on them. This leads to significant challenges when working with complicated models and high-dimensional data.

Our work also loosely connects to the "prototype meta-learning" (Triantafillou et al., 2019; Snell et al., 2017). These studies learn a prototype for every class we need to predict and the final prediction depends on the distances between instances and prototypes. Amortized bayesian prototype meta-learning (Sun et al., 2021) assumes a distribution over class prototypes. This design requires prior knowledge about the classes of tasks and only applies to the classification homogeneous-meta setup.

Another line of related works studies neural approximators of the stochastic process family (Garnelo et al., 2018b; Wang and Van Hoof, 2020; Louizos et al., 2019; Kim et al., 2018). They learn a prior for every task or further use a hierarchical model that learns the instance prior. However, these methods don't share knowledge across tasks. Table S1 compares related lines of works with ours.

Table 1: A summary of datasets and tasks.

| Problems | Tasks | Cardinality | $|\boldsymbol{D}_{\mathcal{T}}^{tr}| \to |\boldsymbol{D}_{\mathcal{T}}^{te}|$ |
|---|---|---|---|
| Regression | 2D regression | $k = 6$ | $10 \to 40$ |
| | Weather prediction | $k > 9000$ | $10 \to 100$ |
| | Image completion | $k = 3$ | $40 \to 784$ |
| Classification | PlainMulti classification | $k = 4$ | 5way 5shot |
| | CelebA binary classification (see Appendix S4) | $k = 1$ | 2way 5shot |

## 3 EXPERIMENTS

Our experiments are designed to answer the following:

**Q1.** Does `ST-MAML` successfully meta-learn from heterogeneous tasks across a variety of applications?
**Q2.** How does `ST-MAML` perform when we have more or less task ambiguity?
**Q3.** How does `ST-MAML` compare to previous heterogeneous meta approaches in terms of accuracy and adaptation?
**Q4.** How does `ST-MAML` perform when applied to a challenging real-world dataset?
To answer **Q1**, we select a wide range of applications in our experiments. We provide a summary of our experimental datasets, and their properties in Table 1 .

Table 2: Prediction error with 95% confidence interval on 2D regression tasks.

| Model | MAML | MetaSGD | BMAML | MMAML | HSMAML | ST-MAML | ST-MAML w/o aug | ST-MAML w/o tarilor |
|-------|------|---------|-------|-------|--------|---------|-----------------|---------------------|
| MSE | $2.29 \pm 0.16$ | $2.91 \pm 0.23$ | $1.65 \pm 0.10$ | $0.52 \pm 0.04$ | $0.44 \pm 0.03$ | $\mathbf{0.37 \pm 0.04}$ | $0.44 \pm 0.05$ | $0.41 \pm 0.06$ |

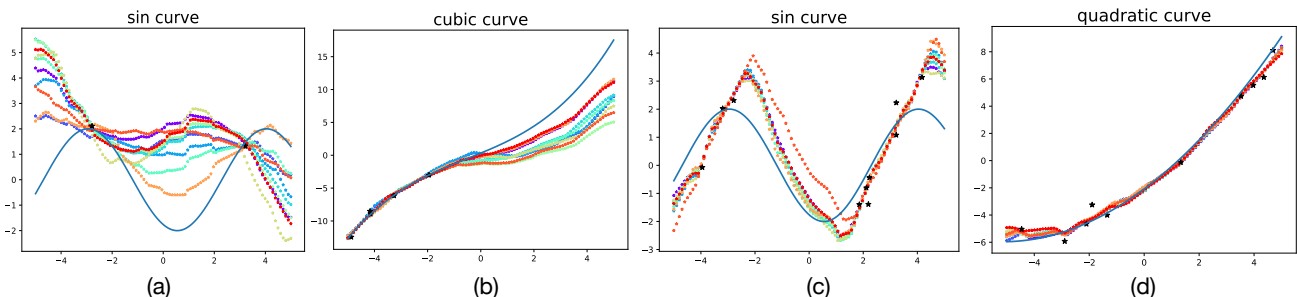

Figure 4: Few-shot 2D regression with various number of training data and noise level. (a) $|\boldsymbol{D}_{\mathcal{T}}^{tr}| = 2, \sigma = 0.3$ (b) $|\boldsymbol{D}_{\mathcal{T}}^{tr}| = 5, \sigma = 0.3$, (c) $|\boldsymbol{D}_{\mathcal{T}}^{tr}| = 10, \sigma = 0.8$, (d) $|\boldsymbol{D}_{\mathcal{T}}^{tr}| = 10, \sigma = 0.1$. Black star represents training data, dashed lines characterize different sampled models, the blue curve is the true mapping. Solutions sampled from ST-MAML span a wide range and stay faithful around annotated data.

We compare against several baselines representing four meta-learning groups: (1) meta-learning methods designed for homogeneous tasks: MAML (Finn et al., 2017) and MetaSGD (Li et al., 2017). (2) Meta-learning methods designed for heterogeneous tasks including MMAML (Vuorio et al., 2019) and HSMAML (Yao et al., 2019). (3) Bayesian meta-learning methods: Bayesian MAML (Yoon et al., 2018), which recasts MAML into the Bayesian framework. (4) Neural processes (NPs) methods (Garnelo et al., 2018a,b). NPs learn a distribution over solutions and are regarded as state-of-the-art methods for small scale meta-learning regression applications.

### 3.1 2D REGRESSION: SIMULATED STUDIES

To answer **Q2**, we generate synthetic heterogeneous regression tasks that come from multiple functional families of curves. We use probabilistic meta-learning models to sample and visualize multiple solutions.

**Setup.** We follow a similar setup as (Yao et al., 2020) to generate 2D regression tasks. The meta-distribution set $\{P_k(\mathcal{T})\}$ consists of 6 function families including *sinusoids, straight line, quadratic, cubic, quadratic surface*, and *ripple* functions. We perturb the output by adding Gaussian noise with standard deviation 0.3 in meta-train tasks. During meta-training, every task is uniformly randomly sampled from one of the 6 function families, and the size of the training set $|\boldsymbol{D}_{\mathcal{T}}^{tr}| = 10$. We adopted mean square error (MSE) to measure prediction accuracy. A detailed description of the setup and model architecture is available in Appendix S5.

**Results, ablations, and analysis.** We train models on around $10,000$ tasks and evaluate it on over $1,000$ newly sampled tasks. The results are summarized in Table 2. We can see clearly ST-MAML outperforms the baselines. To better investigate the contribution of each component, we perform ablation experiments by either removing model tailoring or input variable augmentation. Table 2 shows that both types of task-specific knowledge provide important contributions and their combination gives the best perfor-

mance.

We visualize example curve fits in Figure 4 and Figure S1. During meta-testing, we can decrease the size of training set or increase the noise level such that tasks ambiguity can be more concerning. In Figure S1, all sampled solutions are close to the ground-truth since tasks are less uncertain. Differently, Figure 4 shows that as tasks become more ambiguous, those sampled solutions by ST-MAML tend to span a wider range. However, they stay faithful around those annotated training data. More analysis visualization results can be found in Appendix S5.

### 3.2 HETEROGENEOUS FEW-SHOT CLASSIFICATION

To answer **Q3**, we apply ST-MAML on two common few-shot classification benchmarks from the literature. With space limit, results on CelebA data are in Appendix S4.

**Setup** N-way K-shot classification is a popular setup in few-shot meta-learning (Chen et al., 2019; Ren et al., 2018; Vinyals et al., 2016). The training set of every task consists of $N$ classes with $K$ labeled data in each class. In the benchmark Plain-Multi dataset, each meta-task is sampled from one of four diverse datasets (Yao et al., 2019). We follow the benchmark architecture, including a feature learner using four convolutional blocks. Our ST module takes the input $\boldsymbol{x}$ into two convolutional blocks with 6 channels, and then concatenate the output vector with the target variable into a two-layer MLP to model the mean and variance of $\boldsymbol{Z}_{\mathcal{T}}$.

**Results, ablations, and analysis.** After training on over $50,000$ total tasks, the model is evaluated on $1,000$ tasks for each dataset and the results are summarized in Table 3. The most relevant method is MMAML. It learns a deterministic task embedding with an RNN module and encodes all parameters in both base learner $\boldsymbol{f}_{\boldsymbol{\theta}_b}$ and task learner $\boldsymbol{f}_{\boldsymbol{\theta}_c}$. Our method outperforms it on every dataset. Also, the probabilistic framework enables us to achieve consistently low variance. Note that HSMAML uses the prior knowledge about the number of clusters, which plays an important role

Table 3: 5-way 5-shot classification accuracy with 95% confidence interval on Plain-Multi dataset.

| Settings | Algorithms | Data: Bird | Data: Texture | Data: Aircraft | Data: Fungi |
|---|---|---|---|---|---|
| 5-way 5-shot | MAML | $68.52 \pm 0.79\%$ | $44.56 \pm 0.68\%$ | $66.18 \pm 0.71\%$ | $51.85 \pm 0.85\%$ |
| | MetaSGD | $67.87 \pm 0.74\%$ | $45.49 \pm 0.68\%$ | $66.84 \pm 0.70\%$ | $52.51 \pm 0.81\%$ |
| | BMAML | $69.01 \pm 0.74\%$ | $46.06 \pm 0.69\%$ | $65.74 \pm 0.67\%$ | $52.43 \pm 0.84\%$ |
| | MMAML | $70.49 \pm 0.76\%$ | $45.89 \pm 0.69\%$ | $67.31 \pm 0.68\%$ | $53.96 \pm 0.82\%$ |
| | HSMAML | $\mathbf{71.68 \pm 0.73}\%$ | $\mathbf{48.08 \pm 0.69}\%$ | $\mathbf{73.49 \pm 0.68}\%$ | $\mathbf{56.32 \pm 0.80}\%$ |
| | ST-MAML | $\mathbf{72.49 \pm 0.53}\%$ | $46.51 \pm 0.42\%$ | $\mathbf{72.64 \pm 0.44}\%$ | $\mathbf{55.29 \pm 0.57}\%$ |
| | ST-MAML (w/o aug) | $71.49 \pm 0.55\%$ | $\mathbf{47.17 \pm 0.44}\%$ | $71.62 \pm 0.43\%$ | $54.91 \pm 0.56\%$ |
| | ST-MAML (w/o tailor) | $71.48 \pm 0.55\%$ | $46.07 \pm 0.40\%$ | $70.46 \pm 0.44\%$ | $54.59 \pm 0.56\%$ |

with respect to the final accuracy. Our `ST-MAML` does not reply upon such prior and achieves lower variance and similar performance than HSMAML. We again run two ablated versions of the proposed `ST-MAML`, and compare it against the full version. The combination of input augmentation and model tailoring yields the best results and is most capable of confronting task-heterogeneity.

### 3.3 REAL-WORLD TEMPERATURE PREDICTION

Now we answer **Q4** by applying `ST-MAML` to a challenging regression problem using real-world data.

**Setup.** The NOAA Global Surface Summary of the Day (GSOD) dataset contains daily weather data from thousands of stations around the world. Each task is created by sampling data points from (station, year) pairs. Each sample takes in one date of the year along with 15 weather features such as wind speed, station elevation, precipitation, fog, air pressure, etc for that date. It then learns to predict the average temperature in Fahrenheit on that day. We remove important information like the weather station number, name, latitude, and longitude. Hiding the station information in this way creates a highly heterogeneous problem where each station generates its own task distribution. The model sees 10 days of labeled temperature data before predicting the temperature on 100 test days. More technical details can be found in Appendix S5.

**Results and analysis.** After 100 epochs of training on approximately $42,000$ unique (station, year) tasks, we evaluate the model on a test set of $1,000$ (station, year) pairs. The results are summarized in Table 4. `ST-MAML` predictions are approximately $40\%$ more accurate than MAML. MetaSGD, designed for homogeneous meta-learning, achieves low accuracy because the globally learned learning knowledge hurts the model's generalization to unseen tasks from different distributions. This is consistent with our assumption that incorporating task-specific knowledge into the model can help solve the task-heterogeneous challenge. We also perform ablation experiments in Table 4. Both tailored initialization and augmented features outperform the baselines, and they combine for further improvement. Figure S2 pro-

vides a visualization of trained `ST-MAML` on the NOAA-GSOD temperature prediction task.

### 3.4 HETEROGENEOUS IMAGE COMPLETION

While we have already demonstrated `ST-MAML` on regression and classification tasks, we continue to answer **Q1** by expanding to image completion, which is a popular small scale meta-learning task.

**Setup.** In our heterogeneous image completion application, the meta distribution set $\{P_k(\mathcal{T})\} = \{\text{MNSIT}, \text{FMNIST}, \text{KMNIST}\}$. Every task contains one image of size $28 \times 28$ sampled randomly from one of the three dataset distributions. In meta-training, 40 pixels are observed for every image, thus, $|\boldsymbol{D}_{\mathcal{T}}^{tr}| = 40$. We use coordinates as inputs and pixel value as the target variable. Each image completion can be interpreted as a meta-learning task which generalizes the knowledge from a limited training set $|\boldsymbol{D}_{\mathcal{T}}^{tr}| = 40$ to the entire image of size $|\boldsymbol{D}_{\mathcal{T}}^{te}| = 784$. Architecture details can be found in Appendix S5.

**Baselines, results and analysis.** The described setup is a benchmark task for Neural processes (Garnelo et al., 2018a,b). Thus, we compare our proposed `ST-MAML` with neural processes (NP) (Garnelo et al., 2018b) and conditional neural processes (CNP) (Garnelo et al., 2018a) which is a deterministic NP. The numerical comparison is shown in Table 5. `ST-MAML` achieves higher completion precision compared with NP and CNP. We leave out the variance for all methods because the difference is insignificant.

Image completion task can be highly ambiguous, because there exist multiple full image choices that could explain the pattern of a handful of observed pixels, especially for gray images. Uncertainty arises on three levels: the inter-class level, inter-distribution level, and cross-distribution level. `ST-MAML` can capture more potential truths by learning a distribution of possibilities rather than a unique mapping. We visualize observations and their completions in Figure 5. Interestingly, when we compare the two half-rows describing image completion for a button-up shirt image, the half-row with more pixels observed during meta-testing,

Table 4: 10-Shot temperature prediction. Mean square losses are averaged across over 1,000 sampled test tasks.

| Model | MAML | MetaSGD | ST-MAML | ST-MAML (w/o aug) | ST-MAML (w/o tailor) |
|-------|------|---------|---------|-------------------|----------------------|
| MSE | $141.43 \pm 9.33$ | $291.42 \pm 14.89$ | $\mathbf{86.56 \pm 4.89}$ | $100.27 \pm 5.87$ | $106.37 \pm 5.77$ |

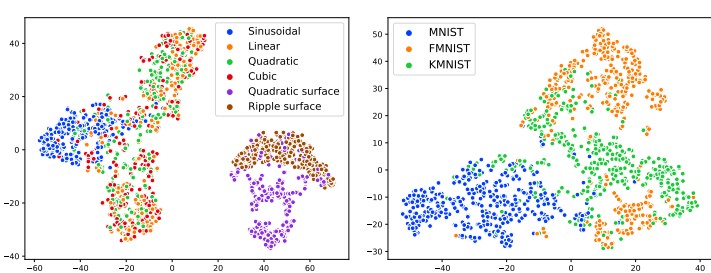

Figure 5: Visualization of completed images. First column contains original images, second column shows the observations which contains 8 annotated pixels (left) and 40 annotated pixels (right). The unobserved pixels have been colored blue for better clarity. The remaining columns correspond to 4 different sampled solutions (completed images) given observations.

Figure 6: t-SNE plots of gate vectors for tasks randomly sampled from the meta-distributions of synthetic regression (left) and image completion (right). Best view in color.

its task is less ambiguous. Therefore, its completed images are closer to the original image. This reflects one merit of ST-MAML : its set operations allows ST-MAML to learn from any size of the training set during meta-testing.

Table 5: Image completion accuracy. Binary cross entropy values are averaged across 300 test tasks.

| Model | NP | CNP | ST-MAML (deter) | ST-MAML |
|-------|-----|-----|-----------------|---------|
| BCE | 0.302 | 0.358 | 0.272 | **0.268** |

### 3.5 VISUALIZATION OF GATE VECTORS $\mathbf{w}_{gate}$.

As noted in Section (2.4), gate vector $\boldsymbol{w}_{gate}$ (Eq 12), which originates from stochastic task variable $\boldsymbol{Z}_{\mathcal{T}}$, translates global initialization $\boldsymbol{\theta}$ to task-specific initialization $\boldsymbol{\theta}_{\mathcal{T}}^0$. Thus, we hypothesis patterns of gate vectors contain information about the relationships between similar tasks. To gain insights into the tasks' gate vectors $\boldsymbol{w}_{gate}$, we visualize sampled vectors on two applications: 2D regression and image completion. For both applications, we sample 200 tasks from each $P_k(\mathcal{T})$, and visualize their gate vectors $\boldsymbol{w}_{gate}$ using a t-SNE plot (Van der Maaten and Hinton, 2008). The visualizations are shown in Figure 6. Gate vectors of tasks from the same distribution (shown as same color points) are clustered on t-SNE plots while tasks from very distinct distributions are further away. For instance, In Figure 6 left, sinusoidal regression tasks (blue) sit far away from ripple surface tasks (brown). These observations can be seen as evidence of the task identification capability of ST-MAML

. Furthermore, tasks from similar distributions may entangle (Figure 6 left, linear, quadratic, and cubic regression tasks). The uncertain identity of similar tasks justifies the representation of task information as stochastic variables.

## 4   CONCLUSION

Task heterogeneity is one critical challenge in meta-learning. Most meta-learning methods assign the same initialization to every task and fail to handle task heterogeneity. ST-MAML encodes tasks using a stochastic task module with set-based operations for permutation-invariance. The probabilistic framework allows us to learn a distribution of solutions for ambiguous tasks and recover better potential task identities. This stochastic task design allows for customizing global knowledge with a learned stochastic task distribution. Empirically, we design extensive experiments on various applications and show that ST-MAML provides an effective way to learn from diverse and ambiguous tasks. As next step, we plan to add domain generalization during meta-testing to enhance our work.

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
