# OpenReview forum: "ST-MAML: A Stochastic-Task based Method for Task-Heterogeneous Meta-Learning"
_auai.org/UAI/2022/Conference — UAI 2022 Poster_

### Official Review · Reviewer_52uJ · 2022-04-13

**Q2(1) Originality/Novelty:** 3
**Q2(2) Significance/Impact:** 2
**Q2(3) Correctness/Technical Quality:** 3
**Q2(6) Clarity Of Writing:** 2
**Q6 Overall Score:** 5
**Q8 Confidence In Your Score:** 4

**Q1 Summary And Contributions:**

This paper tries to empower MAML in scenarios with heterogeneous tasks. A stochastic neural network module is proposed to summarize each task with a stochastic representation. The stochastic task (ST) strategy learns a distribution of task solutions and allows the meta-model to self-adapt. Experiments on some benchmark datasets show the effectiveness of the proposed method.

**Q2 Assessment Of The Paper:**

More detailed information regarding each of these aspects is given below:

**Q2(4) Quality Of Experiments (Optional):**

2: Fair: The experimental evaluation is weak: important baselines are missing, or the results do not adequately support the main claims.

**Q2(5) Reproducibility:**

2: Fair: Key resources (e.g., proofs, code, data) are unavailable but key details (e.g., proof sketches, experimental setup) are sufficiently well-described for an expert to confidently reproduce the main results.

**Q3 Main Strengths:**

1. It is important and challenging to consider the heterogeneity of tasks in meta learning. This paper focuses on a novel and meaningful direction in meta learning.
2. The proposed method on modeling the ambiguity of tasks in stochastic way is reasonable.


**Q4 Main Weakness:**

1. The main weakness lies in the experiments. The datasets used in the experiments are fairly simple and small-scaled. The heterogeneity of the tasks is not that apparent.


**Q5 Detailed Comments To The Authors:**

The experiments are not convincing enough. Besides the heterogeneity of the datasets is not apparent.  In the ablation study, why two modules cannot bring big improvement separately while the improvement when combining these two is more significant?

**Q7 Justification For Your Score:**

Novel and challenging issue in meta learning;
Unconvincing experiments.

**Q9 Complying With Reviewing Instructions:**

1: Yes.

---

### Official Review · Reviewer_pGK1 · 2022-04-22

**Q2(1) Originality/Novelty:** 3
**Q2(2) Significance/Impact:** 2
**Q2(3) Correctness/Technical Quality:** 3
**Q2(6) Clarity Of Writing:** 3
**Q6 Overall Score:** 5
**Q8 Confidence In Your Score:** 3

**Q1 Summary And Contributions:**

This paper suggests an optimization-based meta-learning strategy, which assumes the presence of heterogeneous tasks. Similar to model-agnostic meta-learning, the proposed framework is to quickly adopt to a new task. The authors demonstrate the effectiveness of the proposed method in diverse experimental circumstances such as simple regression, few-shot classification, and image completion.

**Q2 Assessment Of The Paper:**

More detailed information regarding each of these aspects is given below:

**Q2(4) Quality Of Experiments (Optional):**

2: Fair: The experimental evaluation is weak: important baselines are missing, or the results do not adequately support the main claims.

**Q2(5) Reproducibility:**

3: Good: Key resources (e.g., proofs, code, data) are available and key details (e.g., proofs, experimental setup) are sufficiently well-described for competent researchers to confidently reproduce the main results.

**Q3 Main Strengths:**

+ It is well-written.
+ The proposed method seems sound.

**Q4 Main Weakness:**

- The scheme used in this paper is not the state-of-the-art approach in few-shot learning.
- The circumstances used in this paper is somewhat small.

**Q5 Detailed Comments To The Authors:**

Since the line of the optimization-based meta-learning strategies is not the state-of-the-art approach nowadays, the contributions of this paper are somewhat degraded. However, basically, the contributions are novel and the proposed method is sound.

I am curious how the authors define the terminology of heterogeneous tasks. It should be defined more formally.

It is a minor issue, but quote marks should be like ``SOMETHING'', instead of "SOMETHING".

**Q7 Justification For Your Score:**

Based on the comments described above, I would like to recommend borderline accept for this paper.

**Q9 Complying With Reviewing Instructions:**

1: Yes.

---

### Official Review · Reviewer_4XbG · 2022-04-23

**Q2(1) Originality/Novelty:** 2
**Q2(2) Significance/Impact:** 2
**Q2(3) Correctness/Technical Quality:** 3
**Q2(6) Clarity Of Writing:** 3
**Q6 Overall Score:** 6
**Q8 Confidence In Your Score:** 3

**Q1 Summary And Contributions:**

This paper proposes a novel method, ST-MAML, that empowers model-agnostic meta-learning to learn from multiple task distributions. ST-MAML encodes tasks using a stochastic neural network module, that summarizes every task with a stochastic representation. The proposed Stochastic Task (ST) strategy learns a distribution of solutions for an ambiguous task and allows a meta-model to self-adapt to the current task. ST-MAML also propagates the task representation to enhance input variable encodings.

**Q2 Assessment Of The Paper:**

More detailed information regarding each of these aspects is given below:

**Q2(4) Quality Of Experiments (Optional):**

3: Good: The experimental evaluation is adequate, and the results convincingly support the main claims.

**Q2(5) Reproducibility:**

3: Good: Key resources (e.g., proofs, code, data) are available and key details (e.g., proofs, experimental setup) are sufficiently well-described for competent researchers to confidently reproduce the main results.

**Q3 Main Strengths:**

(1) The motivation and preparatory work of the article are sufficient, and the current related work is described in detail.
(2) The structure of the article is very clear, and the model diagram of each part can intuitively describe the work of the module. These include that the motivation for the article is clearly expressed in Figure 1, and the experimental part is in the form of questions and answers.
(3) The Theoretical Analysis of ST-MAML is also clear. In addition, the article also verified the effectiveness of the ST-MAMAL method under real-world data.


**Q4 Main Weakness:**

(1) Taking Figure 1 as an example, if the new meta-test task does not belong to these three possible function families, will it have a better effect?
(2) ST-MAML propagate task representation Z_T into encoding augmented feature representations and denote as H_T, encoding augmented feature representations should be described in detail, including which data and which parts.
(3) To optimize for the intractable likelihood defined in Equation (15), the authors choose to maximize its lower bound of evidence (a.k.a ELBO), please provide a theoretical rationale for choosing it.


**Q5 Detailed Comments To The Authors:**

(1)	There is a lack of some experiments for parameter tuning in the experiment. Do different data-related parameters take the same value?
(2)	The lack of efficiency comparison with other comparison methods in the experiment increases the adaptive process. Is the running time too high?
(3)	Contributions should be listed in the article.


**Q7 Justification For Your Score:**

The most important part to me is the motivation of the proposed method and the novelty of the proposed method. The main advantage of this paper is that the structural modules are clear, but the specific module descriptions need to be more comprehensive. Some related symbols still need to be described.

**Q9 Complying With Reviewing Instructions:**

1: Yes.

---

### Official Review · Reviewer_HXa7 · 2022-04-24

**Q2(1) Originality/Novelty:** 3
**Q2(2) Significance/Impact:** 2
**Q2(3) Correctness/Technical Quality:** 3
**Q2(6) Clarity Of Writing:** 3
**Q6 Overall Score:** 6
**Q8 Confidence In Your Score:** 4

**Q1 Summary And Contributions:**

This paper aims to deal with tasks from multiple distributions, which belongs to the problem of task heterogeneity in meta-learning. The authors propose a novel method, ST-MAML, which empowers the prior work model agnostic meta-learning (MAML) by encoding tasks using a stochastic neural network module. Finally, they perform ST-MAML on various tasks and validate its effectiveness.

**Q2 Assessment Of The Paper:**

More detailed information regarding each of these aspects is given below:

**Q2(4) Quality Of Experiments (Optional):**

3: Good: The experimental evaluation is adequate, and the results convincingly support the main claims.

**Q2(5) Reproducibility:**

3: Good: Key resources (e.g., proofs, code, data) are available and key details (e.g., proofs, experimental setup) are sufficiently well-described for competent researchers to confidently reproduce the main results.

**Q3 Main Strengths:**

1.The idea of addressing task heterogeneity from the perspective of distribution is novel in the community of meta-learning, which makes sense to me.
2.This paper is theoretically sound due to a clear derivation of the evidence bound (ELBO) in Section 2. The following reasoning, including optimization and statistical analysis, is standard and rigorous.
3.The work of this paper is solid due to its conducting experiments on various tasks, such as two few-shot image classification tasks, one curve regression benchmark, one image completion problem, and a real-world temperature prediction application.

**Q4 Main Weakness:**

1.In the final paragraph of Section 3.1, the authors mainly focus on the analysis of Figure 7 but Figure 7 is in the Supplementary Material. And Figure 4 is also a little ambiguous.
2.In Section 3, the experiment part, the ablation study is almost performed in every experiment by either removing model tailoring or input variable augmentation. The question is whether model tailoring and input variable augmentation are the important parts of the proposed method. If they are not, it is not necessary to conduct such ablation study.
3.In Section 3.4, the sentence “This reflects one merit of ST-MAML...” indicates that the experiment here may be interesting and important. Some more discussion should be made here.

**Q5 Detailed Comments To The Authors:**

1.In Section 1, the last paragraph can be organized more logically to emphasize the point and motivation the authors want to convey.
2.The caption of Table 4 is too brief to reflect its content.
3.Some figures, such as Figure 4, need more explanations.

**Q7 Justification For Your Score:**

The paper proposes a novel method to address task heterogeneity in meta-learning by handling tasks from multiple distributions. In spite of some confusion, the overall theoretical derivation and experiments are well done.

**Q9 Complying With Reviewing Instructions:**

1: Yes.

---

### Decision · Program_Chairs · 2022-05-15

**Decision:**

Accept (Poster)

**Comment:**

Meta Review: This paper proposes a novel method, ST-MAML, that empowers model-agnostic meta-learning to learn from multiple task distributions. The method encodes tasks using a stochastic neural network module, that summarizes every task with a stochastic representation. The proposed Stochastic Task strategy learns a distribution of solutions for an ambiguous task and allows a meta-model to self-adapt to the current task. ST-MAML also propagates the task representation to enhance input variable encodings. Finally, the authors perform ST-MAML on various tasks and demonstrate its effectiveness. Although the results are not the SoTA compared with other methods, all the reviewers agree that the idea makes sense and of novelty. Three reviewers think the theoretical analysis of ST-MAML is also sound. Due to the above reasons, I make the decision to accept the paper.